# Effective Combination Therapy of Angiotensin-II Receptor Blocker and Rifaximin for Hepatic Fibrosis in Rat Model of Nonalcoholic Steatohepatitis

**DOI:** 10.3390/ijms21155589

**Published:** 2020-08-04

**Authors:** Yukihisa Fujinaga, Hideto Kawaratani, Daisuke Kaya, Yuki Tsuji, Takahiro Ozutsumi, Masanori Furukawa, Koh Kitagawa, Shinya Sato, Norihisa Nishimura, Yasuhiko Sawada, Hiroaki Takaya, Kosuke Kaji, Naotaka Shimozato, Kei Moriya, Tadashi Namisaki, Takemi Akahane, Akira Mitoro, Hitoshi Yoshiji

**Affiliations:** Department of Gastroenterology, Nara Medical University, Nara 634-8522, Japan; fujinaga@naramed-u.ac.jp (Y.F.); kayad@naramed-u.ac.jp (D.K.); tsujih@naramed-u.ac.jp (Y.T.); ozutaka@naramed-u.ac.jp (T.O.); furukawa@naramed-u.ac.jp (M.F.); kitagawa@naramed-u.ac.jp (K.K.); shinyasato@naramed-u.ac.jp (S.S.); nishimuran@naramed-u.ac.jp (N.N.); yasuhiko@naramed-u.ac.jp (Y.S.); htky@naramed-u.ac.jp (H.T.); kajik@naramed-u.ac.jp (K.K.); shimozato@naramed-u.ac.jp (N.S.); moriyak@naramed-u.ac.jp (K.M.); tadashin@naramed-u.ac.jp (T.N.); stakemi@naramed-u.ac.jp (T.A.); mitoroak@naramed-u.ac.jp (A.M.); yoshijih@naramed-u.ac.jp (H.Y.)

**Keywords:** hepatic fibrosis, rifaximin, ARB, metabolic syndrome, NASH

## Abstract

The progression of nonalcoholic steatohepatitis (NASH) is complicated. The multiple parallel-hits theory is advocated, which includes adipocytokines, insulin resistance, endotoxins, and oxidative stress. Pathways involving the gut–liver axis also mediate the progression of NASH. Angiotensin-II receptor blockers (ARB) suppress hepatic fibrosis via the activation of hepatic stellate cells (HSCs). Rifaximin, a nonabsorbable antibacterial agent, is used for the treatment of hepatic encephalopathy and has been recently reported to improve intestinal permeability. We examined the inhibitory effects on and mechanism of hepatic fibrogenesis by combining ARB and rifaximin administration. Fischer 344 rats were fed a choline-deficient/l-amino acid-defined (CDAA) diet for 8 weeks to generate the NASH model. The therapeutic effect of combining an ARB and rifaximin was evaluated along with hepatic fibrogenesis, the lipopolysaccharide–Toll-like receptor 4 (TLR4) regulatory cascade, and intestinal barrier function. ARBs had a potent inhibitory effect on hepatic fibrogenesis by suppressing HSC activation and hepatic expression of transforming growth factor-β and TLR4. Rifaximin reduced intestinal permeability by rescuing zonula occludens-1 (ZO-1) disruption induced by the CDAA diet and reduced portal endotoxin. Rifaximin directly affect to ZO-1 expression on intestinal epithelial cells. The combination of an ARB and rifaximin showed a stronger inhibitory effect compared to that conferred by a single agent. ARBs improve hepatic fibrosis by inhibiting HSCs, whereas rifaximin improves hepatic fibrosis by improving intestinal permeability through improving intestinal tight junction proteins (ZO-1). Therefore, the combination of ARBs and rifaximin may be a promising therapy for NASH fibrosis.

## 1. Introduction

In recent years, the number of patients with metabolic syndrome has been increasing worldwide due to lack of satiety or exercise. Metabolic syndrome is a condition that leads to dyslipidemia, high blood glucose, and high blood pressure triggered by visceral fat obesity [1]. Nonalcoholic steatohepatitis (NASH) is considered as liver lesions resulting from metabolic syndrome, as they can develop because of obesity, diabetes, dyslipidemia, and hypertension [2]. The pathogenesis of NASH is complicated. Previously, as a mechanism of NASH, the ‘two hits’ theory was proposed, which suggested that the first hit is fat accumulation in hepatocytes, followed by the second hit involving inflammatory cytokines, oxidative stress or insulin resistance [3]. However, the widely accepted theory is the more recent “multiple parallel hits” hypothesis [4]. Multiple processes abet in the development of NASH progression including insulin resistance, adipocytokines, endotoxins, gut microbiome dysbiosis, and oxidative stress [5]. Bacterial translocation promoted by intestinal bacterial overgrowth and enhanced intestinal permeability is one of the concerning factors. Lipopolysaccharide (LPS), produced by the intestinal microbiota, is transported to the liver via the portal vein [6,7].

Recently, the gut–liver axis has garnered much attention as it is one of the most important links between gut microbiota and extra-intestinal organs. It acts as a communication portal between the intestine and the liver. When the intestinal barrier is damaged and becomes increasingly permeable, the liver is automatically exposed to many enterotoxic factors in addition to intestinal bacteria [8]. They can act on cells of the hepatic innate immune system, like Kupffer cells or astrocytes [9]. 

The gut-liver axis also mediates the NASH progression. LPS are known to exacerbate NASH. Recent reports have showed that LPS/toll-like receptor 4 (TLR4)/NF-κB signaling is critical for the activation of inflammatory pathways associated with NASH [10]. Tight junction proteins (TJPs) are reportedly localized on the apical plasma membrane of epithelial cells and maintain epithelial barrier integrity [11]. Previous studies have reported that decrease in LPS and restoration of intestinal TJP suppress liver fibrosis development in NASH [5,12]. It has been reported that rifaximin improves intestinal permeability, increases TJP expression, and improves tight junction barrier function [13]. One of the main TJPs is zonula occludin-1 (ZO-1), which is expressed on tight junctions of both epithelial and endothelial cells and forms a continuous intercellular barrier between them [14].

Rifaximin (RFX) is minimally absorbed oral antimicrobial agent that is concentrated in the gastrointestinal tract, has broad-spectrum in vitro activity against gram-positive and gram-negative aerobic and anaerobic enteric bacteria, and has a low risk of inducing bacterial resistance [15]. Rifaximin is clinically used in hepatic encephalopathy or travelers’ diarrhea [16,17]. Rifaximin significantly ameliorates hepatic encephalopathy, exerts some anti-inflammatory effects, and reduces endotoxin activity without significantly affecting the composition of the gut microbiome [15,18]. As rifaximin is nonabsorbable, systemic adverse effects are uncommon. Rifaximin may inhibit liver fibrosis by reducing LPS through improving intestinal tight junctions. We previously demonstrated that blockade of angiotensin-II (AT-II) signaling through AT-II type 1 receptor (AT1R) prevents hepatic fibrogenesis in rats [19,20]. The inhibitory effects of angiotensin-II receptor blocker (ARB) mostly accord with the suppression of activated hepatic stellate cells (Ac-HSC) [21]. And more, we demonstrated that AT-II augments LPS-TLR4-NF-κB signaling, which plays a pivotal role in hepatic fibrogenesis, through ARB, and AT1R improves liver fibrogenesis and decreases TLR4-mediated innate immune signaling in Ac-HSC [22,23]. In this study, hepatic fibrogenesis-inhibiting effects and mechanism were examined by combining ARB and rifaximin. Furthermore, we hypothesized that the combination of ARB and rifaximin may be useful in suppressing NASH fibrosis.

## 2. Results

### 2.1. Experimental Findings

The findings at sacrifice in each experimental group are shown in Table 1. Mean body weight in control group was significantly higher than the other groups. There was no difference in body weights between CDAA, ARB, RFX, and ARB + RFX. There is no difference in the mean liver weights of each of the groups (control, CDAA, ARB, RFX, and ARB+RFX). The liver-to-body weight ratio in control group was significantly lower than the other groups (*p* < 0.05).

### 2.2. Effect of ARB and Rifaximin on Liver Fibrosis

We examined the effects of ARB (30 mg/kg of losartan, a clinically comparable dose) and rifaximin (100 mg/kg) on hepatic fibrogenesis using Sirius red staining. As shown in Figure 1, the Sirius red staining of liver tissue showed no fibrosis in the control group; contrastingly, obvious fibrosis was observed in the CDAA group. Sirius red staining revealed the presence of stage 3 hepatic fibrosis in CDAA groups. Sirius red staining was significantly decreased in the ARB (*p* < 0.001) and RFX groups (*p* < 0.001) as compared with the CDAA group, whereas it was further decreased in the ARB + RFX group (*p* < 0.001).

Immunohistochemical staining of αSMA and its mRNA expression levels were analyzed to evaluate the activation of HSCs, which play a pivotal role in hepatic fibrogenesis. Hepatic tissue with positive αSMA staining and αSMA mRNA levels were significantly increased in the CDAA group as compared to the control group (*p* < 0.001), whereas they were decreased in ARB (*p* < 0.001) and RFX (*p* < 0.01) and further decreased in ARB+RFX (*p* < 0.001) (Figure 2).

### 2.3. Effect of ARB and Rifaximin on Intestinal Permeability

Intestinal epithelial permeability is structured by intercellular TJP complexes comprised of various components that include ZO-1 or occludin. To identify the changes in intestinal permeability, we evaluated the effect of ARB and rifaximin on ZO-1 expression. The intestinal expression of ZO-1 was clearly observed on the apical side of the intestinal mucosa in control group (Figure 3a). Compared with control, CDAA showed a statistically significant decrease in ZO-1-positive areas and ZO-1 mRNA (*p* < 0.001). In contrast, ZO-1-positive areas and ZO-1 mRNA levels increased in RFX and ARB + RFX (*p* < 0.001). On the other hand, ARB showed no difference of ZO-1-positive areas and ZO-1 mRNA compared to CDAA (Figure 3).

### 2.4. The Inhibitory Effect of Both ARB and Rifaximin on Portal Endotoxin

As rifaximin decreased intestinal permeability by improving TJP, we studied the effect of ARB and rifaximin on the portal blood levels of endotoxins, particularly LPS which are related to gut-liver axis. As it is difficult to measure portal venous LPS levels, we examined portal concentration of LPS-binding protein (LBP). LBP concentration of the portal vein was significantly elevated in CDAA compared to control, and portal LBP levels decreased in RFX (*p* < 0.01) and ARB + RFX (*p* < 0.01); however, ARB did not change compared to CDAA (Figure 4).

### 2.5. Effect of ARB and Rifaximin on LPS-TLR4 Signaling

We evaluated the effect of both ARB and rifaximin on TLR4 mRNA expression of the liver. Hepatic gene expression of LBP was unchanged in CDAA and ARB, but decreased in the groups administered rifaximin (RFX, and ARB + RFX) (*p* < 0.05) (Figure 5a). The levels of TLR4, NF-κB, and TGF-β were significantly decreased in ARB and RFX, compared to CDAA and the levels were lowest in ARB + RFX (*p* < 0.001) (Figure 5b–d).

ARB does not influence LBP expression, whereas rifaximin reduces LBP expression. ARB and rifaximin both individually reduce the expression of TLR4, NF-κB, and TGF-β, and the combination of ARB and rifaximin therapy synergistically reduces their expression.

### 2.6. Effect of ARB and Rifaximin on Intestinal Epithelial Cell Line

The effect of ARB and rifaximin was examined on the rat intestinal epithelial cell line IEC-6 after stimulation with LPS (2 mg/mL). The expression levels of TLR4, NF-κB, IL-6, and LBP did not differ after ARB and rifaximin administration (Figure 6 and Figure 7). The expression levels of ZO-1 also did not differ after ARB administration (Figure 8a). On the other hand, the expression levels of ZO-1 were increased after rifaximin administration in proportion to the rifaximin concentration (Figure 8b).

## 3. Discussion

We previously reported that ARB inhibits liver fibrosis in an animal model [24]. However, the use of a single clinical agent like ARB to inhibit development of hepatic fibrosis has proven to be challenging in clinical settings [25], so we hypothesized that a combinatorial treatment may improve the outcome in NASH fibrosis. This study was designed to assess the effectiveness of combination of ARB and rifaximin in protection against hepatic fibrosis. In our current study, both ARB and rifaximin showed an antifibrotic effect on the NASH liver. Moreover, the combination of ARB and rifaximin demonstrated a greater antifibrotic effect on the NASH liver compared with either of the agents alone.

Our previous report showed that treatment with ARB effectively ameliorated liver fibrosis which is induced by choline-deficient l-amino acid defined (CDAA) diet fed for 12 weeks [24]. AT-II and vascular endothelial growth factor (VEGF) are strongly concerning in hepatic fibrogenesis. ARB was shown to significantly suppress the development of hepatic fibrosis accompanied by VEGF expression in the liver [24]. The underlying mechanism is that ARB directly inhibits activation of HSCs, and AT-II is important for the upregulation of TLR4 expression through the stimulation of AT1R in Ac-HSCs [22]. AT-II and LPS-TLR4-NF-κB signaling play an important role in developing liver fibrosis by modulating TGF-β1 production, which is a key regulator of hepatic fibrosis which promotes fibrosis through stimulation of HSCs [26]. In our current study, we found that ARB and/or rifaximin treatments reduced hepatic α-SMA, TLR4, NF-κB, and TGF-β levels, and consequently hepatic fibrogenesis (Figure 2c and Figure 5b–d). The combination of ARB and rifaximin caused synergistic reduction in hepatic α -SMA, TLR4, NF-κB, and TGF-β levels.

Rifaximin is a poorly absorbed oral antibiotic used against local enteric bacteria with low risk of adverse effects. Rifaximin is used for patients with hepatic encephalopathy or travelers’ diarrhea, or *Clostridioides difficile* colitis [16,17]. In our previous clinical experiment, rifaximin caused only a slight change in intestinal flora [18]. However, the exact changes of the intestinal flora were not known [13,27]. Previous reports have shown that rifaximin inhibits the activation of NF-κB signaling pathway and downregulates the expression of inflammatory cytokines [27,28]. Rifaximin could reduce the concentrations of TNF-α, IL-6, and endotoxin in patients with decompensated cirrhosis. Rifaximin slows down the fibrogenesis by suppressing the secretion of Kupffer cell-derived TGF-β, concerned with HSC activation through a paracrine mechanism [27]. In our current study, the prevention of the expression of hepatic TLR4, NF-κB, TGF-β by rifaximin highlights its suppressive effect on the TLR4/NF-κB signaling pathway in NASH rat model (Figure 5b–d). Since intestinal tight junctions play a pivotal role in paracellular transport, TJP expression levels directly affect the intestinal permeability of the epithelial barrier [29,30]. TJPs such as ZO-1, are intercellular adhesion molecules that affect the passing of various substances from the intestinal lumen to blood [11,31], and TJPs are indicators of intestinal permeability. NASH patients are associated with increased gut permeability, caused by the disruption of intercellular tight junctions (ZO-1) in the intestine [31]. We previously reported that CDAA diet fed rats showed reduced levels of TJP and accelerated liver fibrosis by activating LPS-TLR4 signaling [5]. In our present study, the expression of ZO-1, was reduced in CDAA group as expected (Figure 3). Its expression levels were improved in the rifaximin-administered groups (RFX, and ARB+RFX). However, ARB does not affect the expression of ZO-1 (ARB group). Our data demonstrated that rifaximin, but not ARB, inhibited CDAA-increased intestinal permeability by increasing the expression of ZO-1.

LPS plays an important role in enhancing NASH inflammation or fibrosis [32]. When intestinal permeability increases, endotoxin flow increases into the liver through the portal vein, and hepatic TLR4 is stimulated. Endotoxins directly stimulate HSCs and induce hepatic fibrosis by TGF-β signaling [23]. As direct measurement of LPS concentration is difficult, we measured LBP concentration, which positively correlates with LPS, in the portal blood [33]. LBP concentration in the portal blood was significantly elevated in CDAA group (Figure 4). This is consistent with previous reports underscoring the role of gut-derived endotoxins in NASH [34]. Rifaximin decreased LBP concentration, while ARB had no significant effect (Figure 4). Hepatic expression levels of LBP mRNA were increased in the CDAA group (Figure 5a) while rifaximin-administered groups (RFX and ARB + RFX) showed reduced levels. On the other hand, ARB had no effect. These results indicated that treatment with rifaximin significantly reduced portal LPS and hepatic LPS expression, which inhibited hepatic TLR4/NF-kB signaling.

To explore the effect of rifaximin on intestinal epithelial cells, IEC-6 cells were examined after LPS stimulation. The In vitro study showed that administration of rifaximin and ARB did not affect the expression of LBP, TNF-α, IL-6, and TLR4 in IEC-6 cells (Figure 6 and Figure 7). On the other hand, the expression levels of ZO-1 were increased after rifaximin administration (Figure 8). This shows that rifaximin regulated intestinal permeability not by directly affecting intestinal epithelial cells, but through tight junction proteins; ZO-1. The mechanism of how rifaximin affects tight junction proteins is not well understood, however. Previous studies showed that rifaximin ameliorated visceral hypersensitivity and reduced intestinal permeability and increased TJP without altering microbiome [27]. Pregnane X receptor (PXR) is a ligand-activated transcriptional factor and nuclear receptor expressed universally along the gut-liver-axis [35]. Rifaximin acts as a gut-specific ligand for human PXR but not mouse or rat PXR [36]. Rifaximin treatment down-regulated the TLR4/NF-κB pathway induced by LPS, through a PXR-dependent mechanism [28]. Treatment of epithelial cells with PXR ligand prevented localization of ZO-1 [37]. So, rifaximin activated PXR may recover the intestinal barrier function by reducing intestinal permeability and gut endotoxin leakage in cirrhotic patients. Rifaximin was reported to affect NF-κB activity via PXR activation in humans [38]. However, PXR is not affected in mice. Therefore, the inhibitory effect of rifaximin on NF-κB mRNA shown in this study may be an underestimation of the potential effect in humans.

In addition, we previously reported the effect of oxidative stress by performing malondialdehyde and 8-OHdG in a CDAA model [39]. Oxidative stress was increased by CDAA, and ARB administration showed decreased oxidative stress compared to CDAA. And rifaximin prevented oxidative stress caused by alcoholic liver injury [40]. So, in some points, both ARB and rifaximin may reduce oxidative stress, which may lead to improve liver fibrosis.

In conclusion, our findings demonstrated that the combination of ARB and rifaximin showed a more potent inhibitory effect on liver fibrogenesis than either agent alone. The schema of this experiment is shown in Figure 9. This shows ARB and RFX prevent the progression of NASH fibrosis through the gut–liver axis. These agents protected against hepatic fibrosis through two different mechanisms: (i) influencing HSCs (activating LPS-TLR4 signaling), and (ii) targeting the intestinal barrier (intestinal tight junction protein), decreasing intestinal permeability and consequently portal endotoxin levels. Overall, a combination ARB and rifaximin may characterize a novel therapy in the prevention of NASH progression for future clinical applications.

## 4. Materials and Methods

### 4.1. Animals and Reagents

In total, 30 male six-weeks-old Fischer 344 (F344) rats (Japan SLC, Hamamatsu, Shizuoka, Japan) were used for the experiment. Rats were housed in a room under controlled temperature (23 ± 3 °C), and light illumination for 12 h a day. Losartan, as an ARB, was purchased from Merck Ltd. (Tokyo, Japan). Rifaximin was kindly provided by ASKA Pharmaceutical Co.; Ltd. (Tokyo, Japan). CDAA diet and a choline-sufficient l-amino acid defined (CSAA) diet were purchased from CLEA Japan Inc. (Tokyo, Japan).

### 4.2. Experimental Design

Animal experiments were carried out for 12 weeks. The animals were divided into five groups (Control, CDAA, ARB, RFX, and ARB + RFX). Rats designated the control group, were fed a CSAA diet and given distilled water ad libitum. The interventional groups were fed a CDAA diet for 12 weeks to create the NASH rat model (CDAA group). ARB and ARB + RFX were orally administered 30 mg/kg body weight of water-dissolved losartan (ARB) every day, and RFX and ARB + RFX were orally administered 100 mg/kg body weight of rifaximin with CDAA diet daily. All animal procedures were performed in accordance with the Declaration of Helsinki and in compliance with the standard recommendations for the proper care and use of laboratory animals. The protocol was approved by the Animal Care and Use Committee of Nara Medical University on May 2017 (Approved No. 12008).

### 4.3. Histology and Immunohistochemistry

In each group, 5-mm-thick liver sections were processed routinely stained for Sirius red staining to evaluate hepatic fibrosis as previously described [41]. Immunohistochemical staining for α-smooth muscle actin (α-SMA) (DAKO, Kyoto, Japan) was performed as previously described [42,43]. To evaluate the stained areas, ImageJ software (National Institutes of Health, Bethesda, MD, USA) were used for semi-quantitative analysis.

### 4.4. Quantitative Real-Time RT-PCR Analysis

1 μg of total RNA was extracted from the frozen liver and intestinal tissues using acidic guanidinium thiocyanate-phenol-chloroform extraction. The mRNA levels of α-SMA, TGF-β, TLR4, NF-κB, TNF- α and LBP in the liver and LBP, TLR4, NF-κB, TGF-β, and ZO-1 in the intestine were measured by Quantitative real time-polymerase chain reaction (qPCR) using the Applied Biosystems StepOne™ Real-Time PCR^®^ (Applied Biosystems, Foster City, CA, USA), as described previously [44]. GAPDH was used as an endogenous control. The primer sequences are listed as below. *Gapdh*, forward 5′-AGC TGA ACG GGA AGC TCA CT -3′ and reverse 5′-CAT TGA GAG CAA TGC CAG CC-3′; α*sma*, forward 5′-ACT GGG ACG ACA TGG AAA AG-3′ and reverse 5′-CAT CTC CAG AGT CCA GCA CA-3′, *Tgf*β*1*, forward 5′-CGG CAG CTG TAC ATT GAC TT-3′ and reverse 5′-AGC GCA CGA TCA TGT TGG AC-3′; *Tlr4*, forward 5′-CCG CTC TGG CAT CAT CTT CA-3′ and reverse 5′-CCC ACT CGA GGT AGG TGT TTC TG-3′; *Nf*-κ*b*, forward 5′-TAC CCT CAG AGG CCA GAA GA-3′ and reverse 5′-TCC TCT CTG TTT CGG TTG CT-3′; *Tnf* α, forward 5′-ACT CCC AGA AAA GCA AGC AA-3′, reverse 5′-CGA GCA GGA ATG AGA AGA GG-3′; *Lbp*, forward 5′-AAC ATC CGG CTG AAC ACC AAG-3′ and reverse 5′-CAA GGA CAG ATT CCC AGG ACT GA-3′; and *Zo-1*, forward 5′-ACC GGA GAA GTT TCG AGA GC-3′ and reverse 5′-CTG TAC TGT GAG GGC AAC GG-3′. The cycling conditions were as follows: Initial holding stage at 95 °C for 20 s. And following stages are 40 cycles of 95 °C for 3 s and 60 °C for 30 s. Finally, the melting curve stage of 3 s at 95 °C and annealing for 30 s at 60 °C are performed.

### 4.5. Immunofluorescence Analysis

5-mm thick slices of formalin-fixed, and paraffin-embedded hepatic specimens were used for all experimental groups. After blocked by 10% normal goat serum with PBS, the tissue slices were incubated with rabbit anti rat polyclonal ZO-1 antibody (1:100; Invitrogen Life Technologies, Carlsbad, CA, USA) at 4 °C for overnight. After that samples were incubated with donkey anti-rabbit secondary antibody conjugated with DyLight 488 fluorochrome (Jackson ImmunoResearch Laboratories, West Grove, PA, USA) for 1 h at room temperature. The nuclei were counterstained with 4′,6-diamidino-2-phenylindole Fluoromount-G (Southern Biotech, Birmingham, AL, USA). Stained slices were inspected using a confocal scanning laser microscope (Leica TCSNT; Leica Microsystems, Wetzlar, Germany) equipped with a digital camera. In total, five images were randomly selected for each sample and using ImageJ Software quantified the staining intensity of the selected images based on a preselected threshold.

### 4.6. Portal Venous LBP Concentration

Portal venous LBP concentration was measured using a commercially available kit (HK503, HyCult Biotechnology, Uden, The Netherlands). Serum samples were initially diluted 1:10 for LBP and assayed according to the instructions of the manufacturer. Absorbance values of LBP were read at wavelength of 450 nm using a microplate spectrophotometer (Multiskan FC, Thermo Fisher Scientific, Waltham, MA, USA). Measurable concentration range of this assay is 1.6 to 100 ng/mL.

### 4.7. In Vitro Study

IEC-6 cells; rat intestinal epithelium cells, were purchased from RIKEN Cell Bank (Ibaraki, Japan). This cell line was cultured by Dulbecco’s modified Eagle’s medium (DMEM) with 5% fetal bovine serum, 4 μg/mL insulin and 1% penicillin/streptomycin in an incubator at 37 °C and 5% CO_2_.

A total of 1 × 10^6^ cells/well were seeded with or without ARB and incubated for 24 h. The IEC-6 cells were randomly divided into four groups: 2 mg/mL LPS group, 2 mg/mL LPS plus 10^−7^ M ARB, 2 mg/mL LPS plus 10^−6^ M ARB, and 2 µg/mL LPS plus 10^−5^ M ARB. Depending upon the experiments, IEC-6 cells were cultured in either 6-well plates. Similarly, a total of 1 × 10^6^ cells/well were seeded with or without rifaximin and incubated for 24 h. The IEC-6 cells were randomly divided into four groups: 2 mg/mL LPS group, 2 mg/mL LPS plus 0.1 mM rifaximin, 2 mg/mL LPS plus 1 mM rifaximin, and 2 mg/mL LPS plus 10 mM rifaximin.

### 4.8. Statistical Analyses

The results are presented as the mean ± standard deviation. And the results were analyzed using Student’s t-test for unpaired data or one-way ANOVA followed by Bonferroni’s multiple-comparison test. SPSS (version 22; IBM, Armonk, NY, USA) was used for statistical analysis. All tests were used two-tailed, and *p*-values < 0.05 were considered statistically significant.

## Figures and Tables

**Figure 1 ijms-21-05589-f001:**
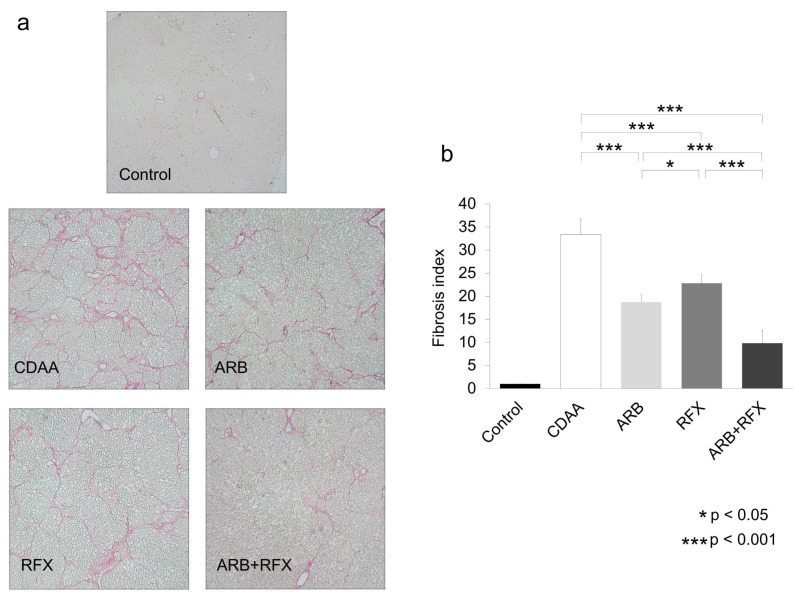
(**a**) Microphotographs of liver sections with Sirius red (magnification; 100 folds). (**b**) Semiquantitative analysis confirmed the histological findings. No fibrosis was observed in the control group. Liver fibrosis was observed in the CDAA group. Monotherapy with an ARB demonstrated a significant inhibitory effect. Monotherapy with RFX demonstrated a significant inhibitory effect. The combination of an ARB and RFX exerted a greater inhibitory effect than that conferred by either monotherapy. Values represent the mean ± SD. * *p* < 0.05, *** *p* < 0.001.

**Figure 2 ijms-21-05589-f002:**
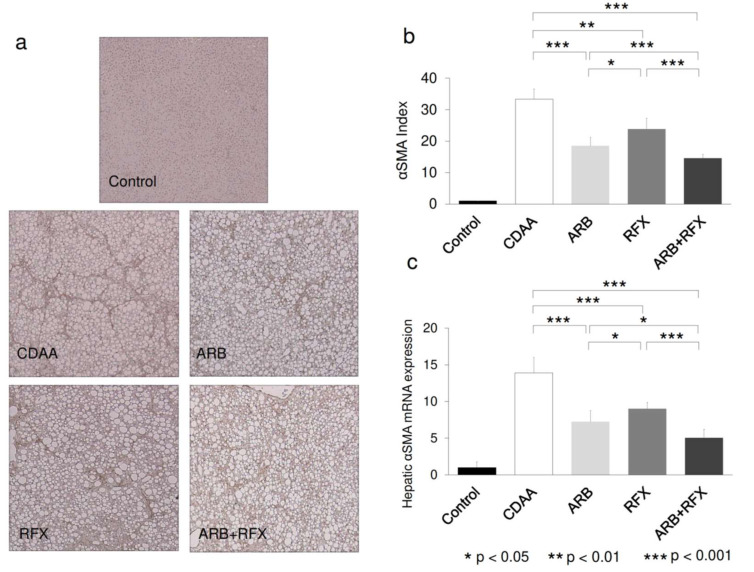
(**a**) Immunohistochemical images of hepatic α-smooth muscle actin (α-SMA) expression. (magnification; 100 folds) (**b**) Semiquantitative analysis of the α-SMA immunohistochemistry was performed using image analysis software. (**c**) hepatic α-SMA mRNA expression. No α-SMA-positive cells were observed in liver sections from the control group. Treatment with either an ARB or RFX resulted in a significant inhibitory effect on hepatic α-SMA mRNA expression compared to that in the CDAA group. The combination of an ARB and RFX exerted a stronger inhibitory effect. Values represent the mean ± SD. * *p* < 0.05, ** *p* < 0.01, *** *p* < 0.001.

**Figure 3 ijms-21-05589-f003:**
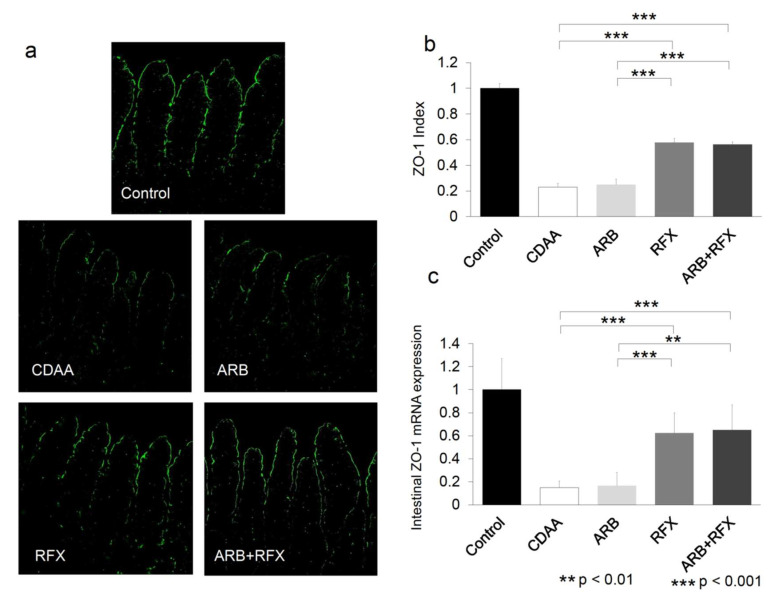
(**a**) Immunofluorescence microphotographs of intestinal ZO-1 expression. (magnification; 200 folds) (**b**) Semi-quantitative analysis of immunofluorescence of ZO-1 expression. (**c**) Semiquantification of RT-PCR results of intestinal ZO-1 expression. ZO-1-positive areas were smaller in the CDAA group than in the CSAA group. Immunofluorescence microscopy was used to evaluate the effect of an ARB and RFX on ZO-1 expression in intestinal tissues. CDAA-induced decreases in ZO-1 expression were significantly increased in the RFX and ARB+RFX groups. No significant increase in intestinal ZO-1 expression was observed in the ARB group. Values represent the mean ± SD. ** *p* < 0.01, *** *p* < 0.001.

**Figure 4 ijms-21-05589-f004:**
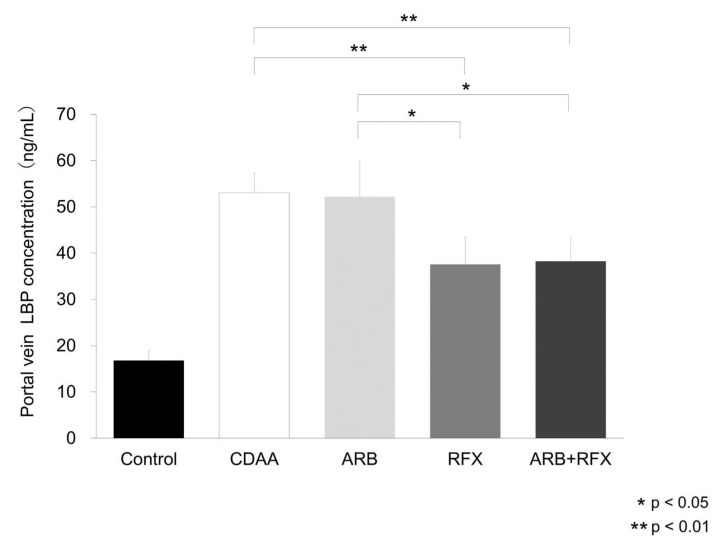
Evaluation of portal vein LPS-binding protein (LBP) concentration. LBP concentrations were significantly increased in the CDAA group compared to those in the control group. CDAA-induced increases in LBP concentration were significantly reduced in the RFX and ARB + RFX groups. No significant reduction in LBP concentration was observed in the ARB group. Values represent the mean ± SD. * *p* < 0.05, ** *p* < 0.01.

**Figure 5 ijms-21-05589-f005:**
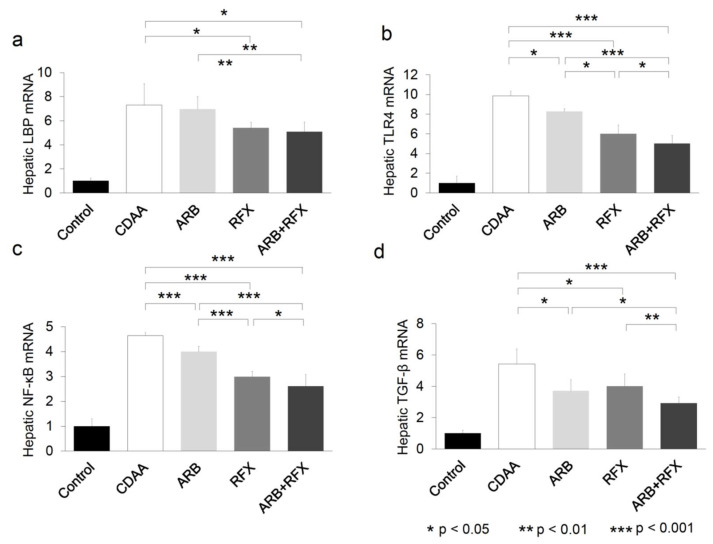
(**a**) Semiquantification of RT-PCR results of hepatic LBP mRNA. (**b**) Semiquantification of RT-PCR results of hepatic TLR4 mRNA. (**c**) Semiquantification of RT-PCR results of hepatic NF-κB mRNA. (**d**) Semiquantification of RT-PCR results of hepatic TGF-β mRNA. Hepatic LBP, TLR4, NF-kB, and TGF-β contents were markedly increased in the CDAA group compared to those in the control group. Treatment with an ARB and RFX significantly suppressed TLR4, NF-kB, and TGF-β mRNA compared to those in the CDAA group. The combination of an ARB and RFX was more effective than with either agent alone. Values represent the mean ± SD. * *p* < 0.05, ** *p* < 0.01, *** *p* < 0.001.

**Figure 6 ijms-21-05589-f006:**
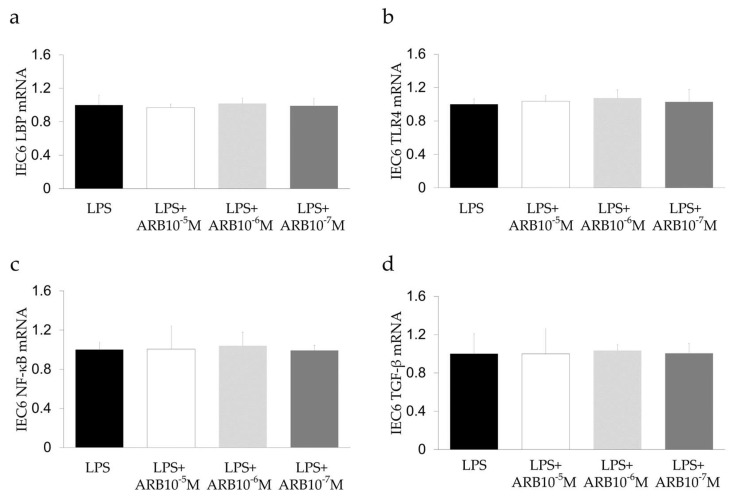
In vitro study of the effect of an ARB on intestinal epithelial cells using IEC-6. (**a**) LBP, (**b**) TLR4, (**c**) NF-kB, and (**d**) TGF-β. ARB did not influence IEC-6.

**Figure 7 ijms-21-05589-f007:**
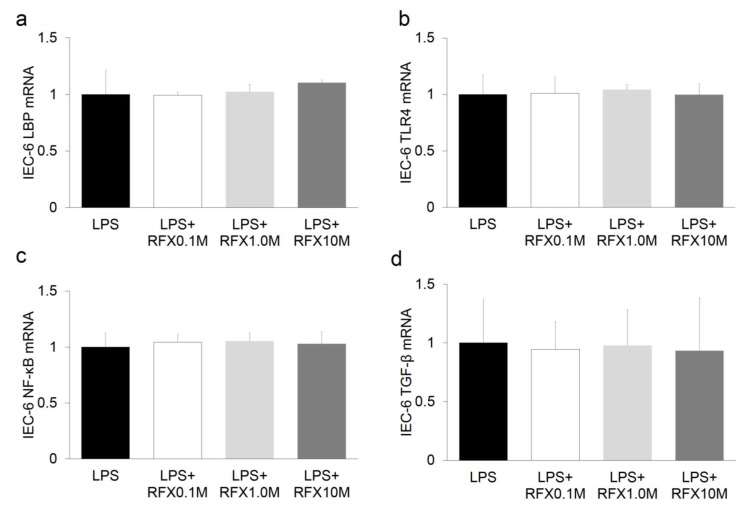
In vitro study of the effect of an RFX on intestinal epithelial cells using IEC-6. (**a**) LBP, (**b**) TLR4, (**c**) NF-kB, and (**d**) TGF-β.RFX did not influence IEC-6.

**Figure 8 ijms-21-05589-f008:**
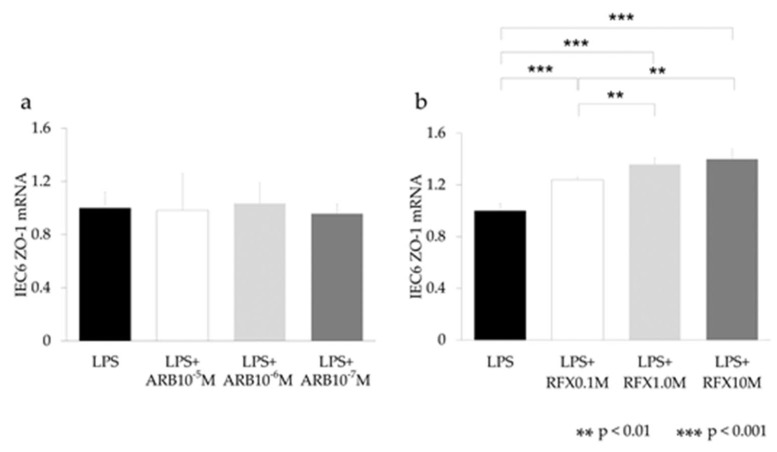
In vitro study of the ZO-1 expression on intestinal epithelial cells using IEC-6. (**a**) ARB, (**b**) RFX. ARB did not increase ZO-1. RFX increased ZO-1 expression in proportion to the LPS concentration. ** *p* < 0.01, *** *p* < 0.001.

**Figure 9 ijms-21-05589-f009:**
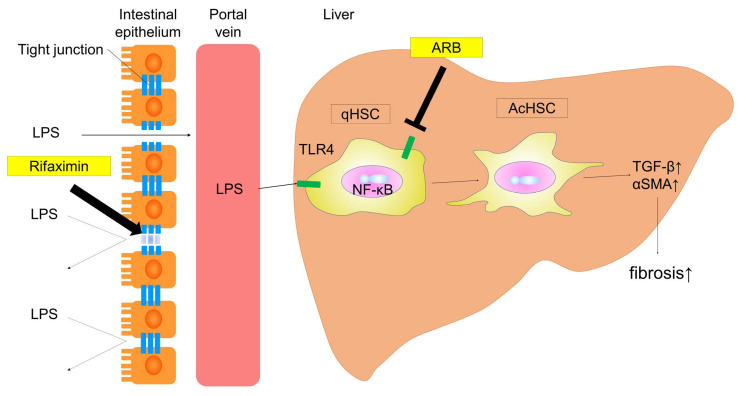
Schema of this experiment. This schema presents the mechanisms by which ARB and RFX prevent the progression of NASH fibrosis through the gut–liver axis. ARB inhibit the activation of hepatic stellate cells by blocking TLR4 and AT-II type 1 receptor, and RFX improves intestinal permeability. Combining ARB and RFX suppresses liver fibrosis through different mechanisms. (Thin arrow; transition to the next. Thick arrow; activation of tight junction protein.)

**Table 1 ijms-21-05589-t001:** Characteristic features of the experimental groups.

Characteristic	Control	CDAA	ARB	RFX	ARB+RFX
Body weight (g)	338.0 ± 16.4	281.7 ± 14.4 ^†^	265.0 ± 17.8 ^††^	271.8 ± 24.0 ^††^	272.2 ± 16.2 ^††^
Liver weight (g)	11.0 ± 0.9	12.4 ± 1.2	11.4 ± 1.4	13.1 ± 1.8	12.1 ± 0.9
Liver weight (%body)	3.2 ± 0.2	4.4 ± 0.2 ^†^	4.3 ± 0.3 ^†^	4.8 ± 0.5 ^†^	4.5 ± 0.2 ^†^

^†^*p* < 0.01 compared with control. ^††^
*p* < 0.001 compared with control.

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
