# Peer review of "Effective Combination Therapy of Angiotensin-II Receptor Blocker and Rifaximin for Hepatic Fibrosis in Rat Model of Nonalcoholic Steatohepatitis"

_ijms, 2020, doi:10.3390/ijms21155589_

Round 1

Reviewer 1 Report

Angiotensin receptor II blocker (ARB) prevents activation of hepatic stellate cell (HSCs) and has been tested whether it can suppress liver fibrosis. Since angiotensin II activates LPS/TLR4/NFkappaB pathway, ARB may suppress fibrosis through suppressing this innate immune signal. Rifaximin an antibiotic for enteric bacteria, reduces the level of LPS through improving tight junctions of the intestinal epithelium. NASH progresses through multiple processes including insulin resistance, adipocytokines, endotoxins, gut microbiome dysbiosis, and oxidative stress and is associated with liver fibrosis. Since activation of LPS/TLR4 pathway in the intestine affects progression of NASH and fibrosis, the authors examined whether the combination of ARB and Rifaximin ameliorates hepatic fibrosis in rat CDAA diet model.

As the authors expected, the combination of ARB and Rifaximin reduced hepatic fibrosis in CDAA model rats more than ARB or Rifaximin alone. The reviewer suggests that the authors explain the following points more clearly.

  1. LPS concentration in the liver judged by LBP was reduced by RFX but not by ARB ( 4). In contrast, expression of TLR4 was downregulated either by ARB or RFX (Fig. 5). Is expression of TLR4 affected by the level of LPS? In that case, why it is downregulated by ARB? The reviewer guesses that the authors focus on HSCs, referring the model shown in Fig. 8. Please explain how ARB and RFX regulate TLR4 signal in HSCs.

  1. In Figure 7, the authors examined expression of the components of TLR4 signals and did not find any effects of RFX. However, RFX has been demonstrated to modulate the barrier function of the intestinal epithelium and actually the authors provided the data ( 3). The authors should examine expression of TJPs in IEC-6 treated with RFX to show whether it transcriptionally regulates the permeability of the intestinal epithelium.  

Author Response

LPS concentration in the liver judged by LBP was reduced by RFX but not by ARB (4). In contrast, expression of TLR4 was downregulated either by ARB or RFX (Fig. 5). Is expression of TLR4 affected by the level of LPS? In that case, why it is downregulated by ARB? The reviewer guesses that the authors focus on HSCs, referring the model shown in Fig. 8. Please explain how ARB and RFX regulate TLR4 signal in HSCs.

>Thank you for your comment. There are AT-II type I receptor and TLR4 in the surface of HSCs. As AT-II augmented the TLR4 expression via AT-II type 1 receptor in the Ac-HSC. So, ARB directly reduce TLR4 expression. However, RFX doesn’t affect to HSCs. The reason why TLR4 expression are downregulated by RFX was portal LBP concentration were reduced by RFX administration. As a result, Ac-HSC are reduced by RFX administration. We added these sentences in the revised manuscript (P10, L201-203, Figure 9 legend)

In Figure 7, the authors examined expression of the components of TLR4 signals and did not find any effects of RFX. However, RFX has been demonstrated to modulate the barrier function of the intestinal epithelium and actually the authors provided the data (3). The authors should examine expression of TJPs in IEC-6 treated with RFX to show whether it transcriptionally regulates the permeability of the intestinal epithelium.

>Thank you for your valuable comment. In accordance with your comment, we evaluated the ZO-1 expression in IEC-6 treated with RFX. ZO-1 expression was increased by RFX treatment. We added the results in the revised manuscript (P8, L174-177) and added Figure 8.

Reviewer 2 Report

The study shows that the combination of Angiotensin-II receptor blocker and Rifaximin improves hepatic fibrosis in rat model of non alcoholic steatohepatitis. This is a good and well-written paper.

It is to underlyne that endotoxin-induced oxidant stress can induce liver injury in NAFLD. So could be of interest to  evaluate the oxidative status, behind the evaluation of sp-NOX2 and urinary 8-iso-PGF-1-alpha in the rats at the end of the therapy.

 In addition, since endotoxin was previously be correlated to NOX2 in NAFLD humans, systemic endotoxemia should be assessed and related to oxidant stress before after therapy in the rats.

Author Response

Reviewer 2

The study shows that the combination of Angiotensin-II receptor blocker and Rifaximin improves hepatic fibrosis in rat model of non-alcoholic steatohepatitis. This is a good and well-written paper.

It is to underline that endotoxin-induced oxidant stress can induce liver injury in NAFLD. So, could be of interest to evaluate the oxidative status, behind the evaluation of sp-NOX2 and urinary 8-iso-PGF-1-alpha in the rats at the end of the therapy.

>Thank you for your comment. Oxidative stress is important in NAFLD. Reviewer suggested to evaluate of sp-NOX2 or urinary 8-iso-PGF-1-alpha, however, we have no antibody of sp-NOX2 or urinary 8-iso-PGF-1-alpha. And, it is difficult to perform additional examination in only five days. Our group previously studied oxidative stress by performing malondialdehyde and 8-OHdG. The data showed increased oxidative stress by CDAA, and ARB showed decreased oxidative stress compared to CDAA. So, our study may show same result. And RFX may also decrease oxidative stress and ARB+RFX may decrease much more. We added these sentences in the revised manuscript (P12, L276-280).

In addition, since endotoxin was previously be correlated to NOX2 in NAFLD humans, systemic endotoxemia should be assessed and related to oxidant stress before after therapy in the rats.

>Thank you for your valuable comment. However, it is difficult to evaluate systemic endotoxin. (We have no peripheral blood of this study; we have only portal venous blood.) There are some reports that peripheral endotoxin is correlate with hepatic inflammation. And hepatic inflammation and oxidative stress are improved by ARB and RFX. So, systemic endotoxin may correlate to hepatic oxidative stress. We try to evaluate in the next experiment.
